# Effects of Botulinum Toxin Therapy on Health-Related Quality of Life Evaluated by the Oromandibular Dystonia Rating Scale

**DOI:** 10.3390/toxins14100656

**Published:** 2022-09-22

**Authors:** Kazuya Yoshida

**Affiliations:** Department of Oral and Maxillofacial Surgery, National Hospital Organization, Kyoto Medical Center 1-1 Mukaihata-cho, Fukakusa, Fushimi-ku, Kyoto 612-8555, Japan; yoshida.kazuya.ut@mail.hosp.go.jp; Tel.: +81-75-641-9161; Fax: +81-75-643-4325

**Keywords:** botulinum toxin therapy, oromandibular dystonia, botulinum neurotoxin, Oromandibular Dystonia Rating Scale (OMDRS), health-related quality of life, non-motor symptom, motor symptom, jaw closing dystonia, tongue dystonia

## Abstract

Oromandibular dystonia (OMD) refers to a focal dystonia in the stomatognathic system. Health-related quality of life (HRQoL) in isolated dystonia is associated with non-motor symptoms such as depression, anxiety, and pain, as well as motor symptoms. To evaluate HRQoL in patients with OMD, the therapeutic effects of botulinum neurotoxin (BoNT) therapy were assessed using a recently developed and validated comprehensive measurement tool called the Oromandibular Dystonia Rating Scale (OMDRS). Altogether, 408 patients (jaw closing dystonia, *n* = 223; tongue (lingual) dystonia, *n* = 86; jaw opening dystonia, *n* = 50; jaw deviation dystonia, *n* = 23; jaw protrusion dystonia, *n* = 13; and lip (labial) dystonia, *n* = 13) were evaluated at baseline and after the end of BoNT therapy or in a stable status. The total OMDRS score reduced significantly from 149.1 to 57.6 (*p* < 0.001). Mean improvement was 63.1%. All examiner-rated subscales (severity, disability, and pain) and patient-rated questionnaire scores (general, eating, speech, cosmetic, social/family life, sleep, annoyance, mood, and psychosocial function) were significantly lower at the endpoint than at baseline (*p* < 0.001). The BoNT injection had a highly positive impact on patient HRQoL, and the OMDRS could evaluate both motor phenomena and non-motor symptoms.

## 1. Introduction

Dystonia is characterized by sustained or intermittent muscle contractions that cause abnormal movements or postures [1]. Oromandibular dystonia (OMD) is a focal dystonia involving the masticatory, lingual, and/or muscles in the stomatognathic system [2,3,4,5,6,7,8,9,10,11,12,13,14]. Based on the direction of abnormal dystonic movements, OMD is classified into six subtypes: jaw closing, jaw opening, tongue (lingual), jaw deviation, jaw protrusion, and lip (labial) dystonia [9,11,12,13,14]. Various combinations of these subtypes have been observed in several cases. A previously reported estimated prevalence of OMD varied from 0.1 to 9.8 per 100,000 persons [11,15,16]. Symptoms related to OMD include masticatory disturbances, limited mouth opening, muscle pain, dysphagia, dysarthria, esthetic problems, and temporomandibular joint dislocation [2,3,4,5,6,7,8,9,10,11,12,13,14]. Some patients exhibit life-threatening features, such as upper airway obstruction due to temporomandibular joint dislocation resulting from severe jaw opening dystonia [17,18] or aspiration pneumonia related to lingual dystonia [19]. These symptoms can result in impaired activities of daily living, social embarrassment, cosmetic disfigurement, absenteeism, unemployment, and a significant impact on a patient’s overall HRQoL [9,13,14]. The symptoms and clinical features may be significantly more variable, critical, and complicated than those accompanied by other focal dystonia, including cervical dystonia or blepharospasm [9]. In 2002, a clinical scoring system for OMD according to subscores for pain, mastication, speech, and discomfort was reported and evaluated in 44 patients with OMD before and after muscle afferent block therapy [2,20]. In 2010, Merz et al. [21] developed and validated the Oromandibular Dystonia Questionnaire (OMDQ-25). In 2019, an oromandibular dystonia screening questionnaire was developed and validated for the differential diagnosis of OMD from other diseases such as temporomandibular disorders, dyskinesia, and functional movement disorders [8]. In 2020, a comprehensive disease-specific Oromandibular Dystonia Rating Scale (OMDRS) was developed and validated [9] (Appendix A). Although several measurement instruments have been used to evaluate various types of dystonia, only a few have been assessed in the clinimetric context [22,23].

Intramuscular injection of botulinum toxin (BoNT) has been successfully applied for OMD as a standard treatment [3,4,5,6,7,10,12,13,14]. Some researchers have attempted to assess the HRQoL in patients with OMD after BoNT therapy [24,25,26,27]. Unfortunately, the number of participants was relatively low, and differences in OMD subtypes were not considered; thus, the reliability remains uncertain. Apart from motor phenomena, there are other non-motor symptoms such as depression, anxiety, sleep problems, and pain in many patients with dystonia [28,29,30,31,32]. Non-motor symptoms are increasingly recognized as important determinants of HRQoL in cervical dystonia [29,30,31,32]. The objective of the present study was to evaluate post-treatment changes in BoNT therapy at the endpoint or in a stable status, particularly in the HRQoL, in patients with OMD using the OMDRS.

## 2. Results

### 2.1. Demographic Data and Results of Treatment 

The demographic characteristics of 408 patients (262 women, 146 men; mean age 52.0 ± 15.6 years (standard deviation [SD])) with OMD are summarized in Table 1. Women (53.8 ± 15.9 years) were significantly (*p* < 0.005, unpaired *t*-test) older than men (48.7 ± 14.4 years). One hundred sixty-eight patients (41.2%) had tardive dystonia. Seventy-three patients (17.9%) had other types of dystonia such as cervical dystonia (10.5%) or blepharospasm (6.1%) (Table 1). However, the symptoms of these types of dystonia were very mild, and the chief complaints were symptoms associated with OMD. 

The main symptoms of the patients were masticatory disturbance (*n* = 146, 35.8%), discomfort, cosmetic problems associated with involuntary movement (*n* = 123, 30.1%), pain (*n* = 78, 19.1%), dysarthria (*n* = 68, 16.7%), and dysphagia (*n* = 41, 10.0%). Evaluation of the OMDRS revealed that such symptoms were more prevalent (cosmetic problem (*n* = 314, 77.0%), masticatory disturbance (*n* = 267, 65.2%), dysarthria (*n* = 230, 56.4%), pain (*n* = 199, 48.8%), and dysphagia (*n* = 159, 38.9%)).

The results of BoNT therapy are shown in Table 2. The mean number of BoNT injection was 5.4 ± 5.0. The main target muscles were the masseter (86.1%), temporalis (49.3%), and medial pterygoid muscles (17.5%) for jaw closing dystonia; the genioglossus and other tongue muscles (100%) and lateral pterygoid muscle (22.1%) for tongue dystonia; the lateral pterygoid muscle (96–100%) for jaw opening, jaw deviation, and jaw protrusion dystonia; and the orbicularis oris (69.2%), risorius (61.5%), and mentalis muscles (38.5%) for lip dystonia (Table 2). The mean number of injected muscles was 3.7 ± 2.0 (Table 2). There were no significant differences in the mean number of injections or injected muscles among the subtypes of OMD.

The mean improvement in the total OMDRS scores was 63.1 ± 18.6%. Fourteen patients (jaw closing, *n* = 3; tongue, *n* = 4; jaw opening, *n* = 6; and jaw deviation, *n* = 1) qualified as partial responders (<30% improvement in total OMDRS score). The proportion of partial responders was significantly higher in jaw opening dystonia (6/50 patients) than in jaw closing dystonia (3/223 patients; *p* < 0.005, Fisher’s exact test). Ten (71.4%) of the 14 partial responders had tardive dystonia, and five patients had other dystonia (cervical dystonia, *n* = 4 and blepharospasm, *n* = 1). The mean follow-up duration from the first visit to the evaluation using the OMDRS for this study was 31.8 ± 28.8 months (Table 2). There were no significant differences in the mean improvement and follow-up among the subtypes of OMD.

Pain scores at baseline were significantly higher in women (10.5 ± 10.8) than in men (6.9 ± 9.5; *p* < 0.005). Examiner-rated scores were also significantly higher in women (27.0 ± 14.7) than in men (23.7 ± 12.7; *p* < 0.005). At the endpoint, women (3.9 ± 5.4) showed significantly higher pain scores than men (2.0 ± 3.5; *p* < 0.005), and significantly higher scores in sleep than men (2.7 ± 3.7 vs. 1.8 ± 2.5; *p* < 0.05).

The adverse effects were temporary regional weakness and tenderness at the injection sites. Approximately 10% of patients with lingual dystonia had mild or transient difficulty in swallowing. Adverse unfavorable events were transient and spontaneously disappeared within 1–2 weeks. No other significant complications were noted.

### 2.2. OMDRS Scores at Baseline and Endpoint

The results of the OMDRS scores (Appendix A) at the baseline and endpoint are summarized in Table 3. Examiner-rated subscales (severity, disability (activities of daily living), and pain) were significantly higher in jaw closing dystonia (27.6 ± 14.3) than in tongue dystonia (21.4 ± 12.3; *p* < 0.01) (Figure 1). Patient-rated questionnaire scores (general, eating, speech, cosmetic, social/family life, sleep, annoyance, mood, and psychosocial function) were significantly higher in jaw opening dystonia (144.1 ± 56.5) than in jaw closing dystonia (115.9 ± 55.0; *p* < 0.01) (Figure 1). OMDRS scores were significantly higher in jaw opening dystonia (170.5 ± 64.1) than in jaw closing dystonia (115.9 ± 62.4; *p* < 0.05) (Figure 1). At the endpoint, patient-rated scores were significantly higher in jaw opening dystonia (65.6 ± 41.8) than in jaw closing dystonia (42.1 ± 29.7; *p* < 0.05) (Figure 1).

The total OMDRS score reduced significantly from 149.1 ± 71.3 to 57.6 ± 40.6 (*p* < 0.001) (Table 3). All examiner-rated subscales (severity, disability, and pain) and patient-rated questionnaires (general, eating, speech, cosmetic, social/family life, sleep, annoyance, mood, and psychosocial functioning) showed significantly lower scores after BoNT therapy (*p* < 0.001) (Table 3).

### 2.3. Differences in Subscales Scores of OMDRS among Subtypes of OMD

Scores of each subscale in each subtype of OMD are shown in Figure 2 (at baseline) and Figure 3 (at the endpoint). 

At baseline, one-way analysis of variance revealed significant differences in disability, pain, eating, speech, social/family life, and annoyance (Figure 2). The disability scores of jaw opening dystonia (12.1 ± 7.1) were significantly higher than those of jaw closing dystonia (8.8 ± 5.7; *p* < 0.005) and tongue dystonia (8.9 ± 4.3; *p* < 0.05). Pain scores of jaw closing dystonia (11.1 ± 10.7) were significantly higher than those of tongue dystonia (5.5 ± 8.9; *p* < 0.001). The eating scores of tongue dystonia (7.3 ± 7.4) were significantly lower than those of jaw opening dystonia (14.5 ± 7.5, *p* < 0.001) and jaw closing dystonia (10.7 ± 7.4, *p* < 0.001), and the eating scores of jaw opening dystonia (14.5 ± 7.5) were significantly higher than those of jaw closing dystonia (10.7 ± 7.4, *p* < 0.05). The speech scores of tongue dystonia (11.1 ± 4.5) were significantly higher than those of jaw closing dystonia (6.8 ± 4.7; *p* < 0.001). The social/family life scores of jaw opening dystonia (11.3 ± 5.6) were significantly higher than those of jaw closing dystonia (7.8 ± 6.0; *p* < 0.005). Annoyance scores of jaw opening (17.7 ± 11.2) were significantly higher than those of jaw closing dystonia (13.8 ± 10.7; *p* < 0.05) and tongue dystonia (16.4 ± 9.4; *p* < 0.05).

At the endpoint, one-way analysis of variance revealed significant differences in disability, eating, speech, social/family life, and psychosocial functioning (Figure 3).

The disability scores of jaw opening dystonia (6.4 ± 6.4) were significantly higher than those of jaw closing dystonia (3.1 ± 3.1, *p* < 0.001) and tongue dystonia (3.4 ± 3.8, *p* < 0.05). The eating scores of jaw opening dystonia (8.0 ± 6.9) were significantly higher than those of tongue dystonia (2.9 ± 4.4; *p* < 0.001) and jaw closing dystonia (4.3 ± 3.7; *p* < 0.001). The speech scores of tongue dystonia (4.6 ± 3.7) were significantly higher than those of jaw closing dystonia (2.7 ± 2.7; *p* < 0.005). Scores of social/family life of jaw opening dystonia (6.2 ± 4.5) were significantly higher than those of jaw closing dystonia (3.4 ± 3.6; *p* < 0.05). The psychosocial functioning scores of jaw opening (11.5 ± 8.2) were significantly higher than those of jaw closing dystonia (6.6 ± 6.7; *p* < 0.05) and tongue dystonia (16.4 ± 9.4; *p* < 0.05).

### 2.4. Differences in OMDRS Scores between Idiopathic and Tardive Cases

The total OMDRS scores at baseline were significantly higher in patients with tardive dystonia (172.1 ± 69.6) than in idiopathic patients (132.3 ± 54.1; *p* < 0.001, unpaired *t*-test). The OMDRS scores at the endpoint also showed significant differences (74.9 ± 41.4 vs. 46.8 ± 36.2, *p* < 0.005). The improvement was significantly lower in tardive cases (58.7 ± 17.8%) than in idiopathic cases (65.9 ± 18.6%; *p* < 0.001). Subjective improvement was also significantly lower in tardive patients (52.3 ± 18.8%) than in idiopathic patients (67.6 ± 17.4%, *p* < 0.001). Improvements in patient-rated scores were significantly lower in tardive patients (57.4 ± 17.8%) than in idiopathic patients (65.5 ± 19.1%, *p* < 0.001). Although improvement of examiner-rated score was lower in tardive cases (60.6 ± 20.1%) than in idiopathic cases (64.6 ± 20.3%), it was not significant (*p* = 0.15).

Most scores of the subscales of the OMDRS were significantly higher (unpaired *t*-test) in patients with tardive dystonia than in those with idiopathic dystonia (Figure 4). At baseline, significant differences were observed in severity (7.9 ± 3.1 vs. 7.3 ± 2.8, *p* < 0.05), disability (10.8 ± 5.7 vs. 7.9 ± 5.2, *p* < 0.001), general (15.1 ± 4.5 vs. 13.6 ± 4.2, *p* < 0.001), eating (12.6 ± 7.3 vs. 8.6 ± 7.5, *p* < 0.001), social/family life (10.9 ± 5.8 vs. 7.2 ± 5.2, *p* < 0.001), sleep (6.4 ± 5.5 vs. 4.1 ± 4.9, *p* < 0.001), annoyance (18.7 ± 7.7 vs. 14.6 ± 7.6, *p* < 0.001), mood (± vs. ±, *p* < 0.001), and psychosocial functioning (19.0 ± 11.0 vs. 12.1 ± 9.1, *p* < 0.001) (Figure 4). At the endpoint, significant differences were observed in disability (4.5 ± 3.8 vs. 3.1 ± 3.8, *p* < 0.01), general (6.5 ± 3.9 vs. 5.3 ± 3.5, *p* < 0.05), social/family life (6.4 ± 4.3 vs. 2.9 ± 3.4, *p* < 0.001), sleep (3.7 ± 3.9 vs. 1.5 ± 2.5, *p* < 0.001), annoyance (11.3 ± 6.8 vs. 5.6 ± 5.4, *p* < 0.001), mood (11.6 ± 6.7 vs. 6.3 ± 5.2, *p* < 0.001), and psychosocial functioning (11.8 ± 7.8 vs. 5.6 ± 6.2, *p* < 0.001) (Figure 4).

## 3. Discussion

This study is the first to report therapeutic effects and post-treatment changes in HRQoL in patients with OMD after BoNT therapy using a comprehensive measurement tool. Differences were compared among the six OMD subtypes in the present study. The OMDRS precisely assessed motor and non-motor features after BoNT therapy, even for each subtype of OMD. Jaw opening dystonia tended to be more severe in disability, eating, and social family/life; jaw closing in pain; and tongue dystonia was severe in speech.

### 3.1. Rating Scales for OMD

Merz et al. [21] developed and validated the OMDQ-25, which consists of five subscales (general, psychosocial, cosmesis, speech, and eating dysfunction). This scale was the first measurement tool to assess the HRQoL in patients with OMD. However, the Movement Disorders Society Task Force on dystonia rating scales did not recommend the OMDQ-25 but merely suggested it because it was used only by the original developers and not by other researchers [22]. Subsequently, the OMDQ-25 was applied in several studies [9,33,34]. OMD symptoms exhibit extremely large individual differences among patients. For instance, jaw closing, jaw opening, and tongue dystonia are completely different in terms of clinical features, affected muscles with abnormal contracture, and direction of abnormal movements. Therefore, the severity subscale should be examined according to OMD subtype. However, if the items are negatively associated within a subscale, the subscale cannot reach sufficient internal consistency, as assessed by Cronbach’s alpha [35]. Therefore, in the OMDRS, only the severity subscale (four items) was assigned five patterns according to the OMD subtype (jaw closing, tongue, jaw opening, jaw deviation (protrusion), and lip dystonia) [9]. The OMDQ-25 is a concise, patient-rated 25-item questionnaire [21]. In contrast, the OMDRS includes a 15-item examiner-rated scale and a 57-item patient-administered questionnaire to comprehensively assess the full spectrum of OMD [9]. Significant correlations between the subscales of the OMDRS and the Short Form-36 Health Survey were reported for several subscales (pain, general, eating, social/family life, sleep, annoyance, and psychosocial functioning) [9]. The OMDRS can be useful for more precise evaluation of disease severity and post-treatment changes in each subtype for both clinical and research purposes.

The patient-administered rating scale, particularly the Cervical Dystonia Impact Profile-58 [36], is more sensitive and reflective of HRQoL than the physician-administered standard recommended measure of cervical dystonia severity in patients after BoNT injection [30]. Four subscales (sleep, annoyance, mood, and psychosocial functioning), a disease-specific questionnaire, and an examiner-rated scale were combined to construct the OMDRS [9] (Appendix A). Therefore, the OMDRS can be sensitive and reflective of non-motor symptoms and health-related quality of life in OMD, as can the Cervical Dystonia Impact Profile-58 in cervical dystonia. Non-motor symptoms are highly prevalent in cervical dystonia [29,30,31,32]. Non-motor symptoms are increasingly recognized as important determinants of HRQoL in various movement disorders [28,29,30,31,32]. Motor improvement after BoNT therapy for cervical dystonia did not correlate with non-motor changes [32]. In BoNT therapy for OMD, special attention should be paid not only to the motor symptoms but also to the non-motor symptoms in order to improve the patient’s HRQoL. 

The chief complaints of the patients (masticatory disturbance, 35.8%; discomfort and cosmetic problems associated with involuntary movement, 30.1%; pain, 19.1%; dysarthria, 16.7%; and dysphagia, 10.0%) were more prevalent (cosmetic problems, 77.0%; masticatory disturbance, 65.2%; dysarthria, 56.4%; pain, 48.8%; and dysphagia, 38.9%) after evaluation of OMDRS. The OMDRS could be used to evaluate very mild symptoms other than the chief complaint in more detail.

### 3.2. Results of This Study

This study was based on real-world clinical results obtained by an OMD specialist. The author has treated several thousands of patients with a variety of movement disorders in the stomatognathic system over the past 30 years. Moreover, the author was able to clinically and scientifically research movement disorders with many neurologists. In the present study, the improvement evaluated by the reduction rate of OMDRS scores was 63.1% without significant adverse effects. Merz et al. [21] reported a reduction rate of 37.6%, Nastasi et al. [33] 15.4%, and Scorr et al. [34] 19.8%, as recalculated by the percentage reduction of the OMDQ-25. The much higher success rate without complications in this study than that of previous studies may be related to the following factors. First, oral surgeons must be more familiar with masticatory and other muscles in the stomatognathic system than neurologists, otorhinolaryngologists, and other medical professionals. A complete understanding of the local anatomy of the stomatognathic system is a prerequisite for target muscle selection and safe injection without complications [12,13,14,37]. The more accurately BoNT is administered to the target muscles, the more likely is the improvement in patient symptoms, and the lower the risk of complications [12,13,14,37]. Second, dental problems such as temporomandibular disorders and bruxism were differentially diagnosed and excluded from BoNT therapy by an expert oral surgeon. Third, patients who were not indicated for BoNT injection, such as those with functional movement disorder [38] or psychiatric disease, were excluded using a muscle afferent block (lidocaine injection) before BoNT therapy [12,13,14,39]. 

Berardelli et al. [40] reported that dystonic movement in patients with cervical dystonia evaluated using the Toronto Torticollis Rating Scale showed significant improvement after 5 years of BoNT therapy (33.4 vs. 26.9). However, neuropsychiatric disorders did not improve at all (65% vs. 64%). Widespread loss of inhibition and pathologically increased plasticity appear to play important roles in the pathophysiology of primary dystonia [41]. Stamelou et al. [42] proposed that the non-motor symptoms of dystonia may be explained by a common pathophysiological deficit that also underlies the motor features. Fifty-one (57.3%) of eighty-nine consecutive patients with various forms of focal dystonia had psychiatric disorders, which started on average 18.4 years before the onset of dystonia, implying that psychiatric features were primary rather than consequences of the movement disorder [43]. In this study, 41.2% of patients had tardive dystonia. Patients with tardive dystonia showed significantly higher scores on the subscales of non-motor symptoms, such as social/family life, sleep, annoyance, mood, and psychosocial functioning (Figure 4). Additional psychiatric treatment may be necessary for patients with mental disorders. A number of patients in the present study lost their jobs because of symptoms related to OMD. However, several patients could return to work after BoNT therapy, and their quality of life and mental status improved considerably. 

Fourteen patients were partial responders (<30% improvement in the total OMDRS score) in this study. Ten patients presented with tardive dystonia. Whether tardive dystonia or primary non-response was the reason for the partial response remains unknown. However, improvement in the patient-rated score was significantly lower in tardive patients (57.4 ± 17.8%) than in idiopathic patients (65.5 ± 19.1%, *p* < 0.001) in this study. Moreover, scores for non-motor symptoms (annoyance, mood, and psychosocial functioning) were significantly (*p* < 0.001) higher in tardive cases than in idiopathic cases (Figure 4). The differences might result in low improvement in the ten patients. The possibility of non-responders was not confirmed in laboratory or clinical tests.

The rate of partial responders was significantly higher for jaw opening dystonia than for jaw closing dystonia. A significantly lower effect of BoNT therapy in jaw opening dystonia than in jaw closing dystonia has been reported [44]. This is considered to be related to the difficulty of accurate injection into the jaw opening muscles (the lateral pterygoid muscle and anterior belly of the digastric muscle) compared to that into the jaw closing muscles (the masseter, temporalis, and medial pterygoid muscles). As the author always injects BoNT under electromyographic (EMG) guidance, BoNT must be administered precisely to the lateral pterygoid or digastric muscles. Moreover, the author used a computer-aided design/computer-assisted manufacture-derived insertion guide into the lateral pterygoid muscles for patients with jaw opening who responded insufficiently to freehand insertion [37,45]. The lateral pterygoid is a muscle involved in mastication and usually has two heads: the inferior (lower) and superior (upper) [46,47,48]. As the bilateral inferior heads contract, the condyle is pulled forward and slightly downward. If the muscle is activated only on one side, the inferior jaw rotates around a vertical axis that runs through the contralateral condyle and is pulled medially to the contralateral side [46,47,48]. The superior and inferior heads are activated alternately during chewing, such that the inferior head contracts during mouth opening, while the superior head relaxes [46,47,48]. Nevertheless, the number of lateral pterygoid muscle heads remains controversial. It is commonly a two-headed muscle, but one-headed or three-headed muscles have also been reported [46,49,50]. Detailed findings of the origins and insertions of an anatomical study suggest that the lateral pterygoid muscle is a single muscle with no clear border, containing fibers in various directions, indicating that a two-head muscle pattern would be indicated by the differences in the convergences of the muscle fibers [49]. In contrast, a recent systematic review reported that the frequency of one-headed lateral pterygoid muscles ranged from 7.7% to 26.7%, two-headed muscles from 61.4% to 91.1%, and three-headed muscles from 4.0% to 35.0% [50]. The difference in efficacy of BoNT therapy between jaw closing and jaw opening dystonia may be associated with the anatomical variability of the lateral pterygoid muscle. However, further studies with larger sample sizes are warranted.

Post-treatment OMDRS scores were significantly correlated with the number of injected muscles and significantly negatively correlated with improvement. In other words, the more affected the muscles, the less effective the BoNT therapy for OMD. As there was no correlation between the degree of improvement and disease duration, it is unlikely that only the exacerbation or expansion of symptoms due to long-term disease duration affected the improvement. Whether the relatively reduced dose of BoNT due to the increased number of muscles affected this improvement was not determined. Therefore, further research is necessary. 

### 3.3. Treatment Modalities of OMD

A recent study reported the crude prevalence of OMD to be 9.8 per 100,000, suggesting that OMD may have an equal or even higher prevalence than cervical dystonia or blepharospasm [11]. Patients with OMD were referred to dentists (70%) or oral surgeons (60%) [51]. However, approximately 90% of patients had not been diagnosed with OMD, and the vast majority of patients had been diagnosed with temporomandibular disorders, bruxism, or psychiatric diseases [11,51]. OMD has been recognized as a rare disease by many neurologists; however, in reality, cases are incorrectly diagnosed [13,14]. 

OMD has been regarded as the most challenging dystonia for neurologists [52]. However, it can be part of the clinical spectrum of various neurological diseases, including Parkinson’s disease, Wilson’s disease, ischemic or hemorrhagic stroke, tumors, infarction, and brain injury [53]. If such diseases have already been diagnosed and treated, OMD must be addressed simultaneously by the attending physicians. Likewise, collaboration with a psychiatrist is required for the treatment of tardive or functional (psychogenic) dystonia [13,14]. Neurological knowledge and experience are indispensable in the diagnosis of neurological diseases. Dental knowledge and experience are required for differential diagnosis of dental and oral conditions. Since dentists and oral surgeons are specialists of the stomatognathic system, they are likely to perform more skillful and accurate injections into the muscles in the oral region than medical professionals [13,14]. Collaboration with medical and dental professionals is important for diagnosis and treatment.

Treatment of OMD must be multimodal and highly individualized for each patient. Treatment options are oral medication [53,54], BoNT [3,4,5,6,7,12,13,14,55,56,57,58,59], muscle afferent block [2,20,60], occlusal splint [61,62], and surgical procedure (coronoidotomy) [63,64,65]. BoNT therapy, namely, chemodenervation with BoNT, is considered the first-line treatment for OMD [6,13,14]. A complete understanding of the local anatomy of the muscles, nerves, and other tissues and accurate injection procedures are prerequisites for BoNT therapy of OMD [13,14,39]. It is important to differentially diagnose the indications for BoNT injections. If an adequate dose of BoNT can be correctly administered to the affected muscles, the symptoms will improve. Previously reported adverse effects of BoNT include temporary regional weakness, tenderness or pain at the injection site, minor discomfort during chewing, asymmetric smiles, muscle atrophy, paresthesia, and difficulty in swallowing [4,56,58,59]. The majority of these side effects are thought to be related to the injection technique and are avoided by accurate knowledge of the local anatomy, precise injection procedures, and optimal dose of BoNT [13,14]. 

The author created websites for involuntary movements, which seem to have received considerable attention from many patients with OMD [51]. A large number of patients worldwide wish to visit our department for treatment of OMD; however, only a few can actually visit because of very high costs, including airfare [51]. Furthermore, overseas travel was prohibited due to coronavirus disease 2019 restrictions, making it impossible for patients from abroad to receive medical examinations. A computer-aided design and manufacturing process was used to develop a needle guide to reliably administer BoNT to the inferior head of the lateral pterygoid muscle [45]. Computed tomography scans with a plaster cast model of the maxilla data can be transmitted over the Internet from anywhere in the world. Telemedicine for OMD using digital technology in the era of coronavirus disease 2019, computer-aided design, and manufacturing of needle guides for lateral pterygoid muscle injection can be applied in response to the demands of overseas patients with OMD [13,14,45].

### 3.4. Limitations and Strengths 

The present study had some limitations that should be considered. An obvious limitation of this retrospective study was its open-label design without a placebo-controlled group. It is likely that the placebo effect influenced the results of this study, particularly some subscales, such as annoyance, mood, and psychosocial functioning. Another weakness of this study was the lack of assessment of depression or anxiety using a dedicated rating scale or specialized examination. Further randomized controlled studies evaluating the mental status in greater detail are necessary.

A strength of this study was that all patients were diagnosed, treated, and evaluated by an OMD specialist to ensure the uniformity of the results. Therefore, inter-clinician or inter-rater differences in diagnosis, treatment, and rating were minimal. Furthermore, because the developers of the rating scale evaluated it themselves, the errors associated with the evaluation or rating might be minimal. Another strength of this study was the number of participants. Approximately 100 patients with various movement disorders of the stomatognathic system visit our department every week. The author launched a website for involuntary movements for both patients and healthcare workers (https://sites.google.com/site/oromandibulardystoniaenglish/, accessed on 1 August 2022) in 2011 [54]. The website has 20 versions in 20 languages. This site has been accessed tens of millions of times, worldwide. Many patients who had already abandoned treatment or further consultation visited our department from all over Japan and abroad [51]. In addition, the author has published many scientific papers and lectures on OMD, not only in neurological but also in dental or maxillofacial surgical societies and patient associations. Efforts to raise awareness about OMD have resulted in an increasing number of referrals from neurologists, neurosurgeons, dentists, oral surgeons, and otorhinolaryngologists. 

### 3.5. Future Directions

BoNT injections have been commonly used for OMD. However, the available evidence is insufficient. An early double-blind, placebo-controlled study of BoNT treatment for cranial–cervical dystonia in 10 patients with oromandibular–cervical dystonia [55] was published, and a recent pilot single-blind study evaluated BoNT dosing and efficacy in 18 patients with OMD [34]. Unfortunately, the small number of patients, low improvement, and high frequency of side effects in these studies might limit the conclusions of BoNT therapy for OMD as an effective treatment choice. However, most other studies on BoNT therapy for OMD were based on retrospective chart reviews. Meanwhile, randomized controlled trials have already been reported for bruxism or temporomandibular disorders, which have symptoms similar to jaw closing dystonia and involuntary jaw closing [66,67,68]. This is likely because bruxism and temporal disorders have a higher prevalence than OMD and there are many more clinicians and researchers. In future, randomized controlled trials with a higher level of evidence should be conducted for OMD.

Although BoNT is frequently used for treatment in many countries, it has not been officially approved for OMD treatment. Well-designed, randomized, controlled trials with larger sample sizes and longer follow-up periods are required to determine the therapeutic efficacy, optimal dose, duration of effect, adverse effects, brand-specific differences, and definite indications, and to establish a protocol for BoNT therapy [13]. However, the presence of disabilities in patients with OMD places constraints on the traditional placebo–control trial design [34]. The apparent lack of effect in the control group may have led to substantial dropouts and compromised the reliability of the statistical analyses. Patients seeking OMD specialists visit from very long distances with very high expectations, making the formation of a control group ethically difficult [13]. However, such randomized controlled studies are indispensable for the official approval of BoNT and evidence-based medicine. The author, along with neurologists specializing in dystonia, is currently planning a randomized, double-blind, placebo-controlled study of BoNT therapy for OMD to seek official approval for BoNT in Japan. We hope to report on evidence-based data in the near future.

A common treatment strategy for dystonia concentrates only on physical symptoms. Nevertheless, HRQoL in isolated dystonia is strongly related to non-motor symptoms and less associated with motor symptoms [31]. Very little attention has been paid to non-motor symptoms in patients with OMD. Comprehensive treatment for OMD should address both physical and mental aspects. Some patients with OMD may require additional mental health care. For this purpose, collaboration with psychiatrists may be necessary, indicating the importance of dental and medical multidisciplinary team approaches.

## 4. Conclusions

BoNT therapy is very effective and safe for OMD when properly diagnosed and administered precisely. The OMDRS can be used to comprehensively evaluate therapeutic effects and HRQoL.

## 5. Materials and Methods

### 5.1. Participants

Four hundred and eight patients (262 women and 146 men; mean age, 52.0 ± 15.6 years) with OMD, who visited our department from January 2016 to January 2021, were enrolled in this retrospective study (Table 1). OMD was differentially diagnosed based on the characteristic phenomenology of focal dystonia, such as stereotypy, task specificity, sensory tricks, overflow phenomenon, morning benefit, co-contraction, and EMG findings, as described in detail previously [8,9,11]. The main symptoms of the patients were masticatory disturbance (*n* = 146, 35.8%), discomfort and cosmetic problems associated with involuntary movement (*n* = 123, 30.1%), pain (*n* = 78, 19.1%), dysarthria (*n* = 68, 16.7%), and dysphagia (*n* = 41, 10.0%).

The inclusion criteria were as follows: (1) age over 18 years, (2) ability to be evaluated for OMDRS by interview or questionnaire, and (3) follow-up of 12 months or longer. The exclusion criteria were as follows: (1) generalized, functional [38], or significant dystonia-related neurological diseases in other body regions; (2) history of surgical procedures such as coronoidotomy [63,64] or deep brain stimulation; (3) good response to other therapies such as oral medicine, mouthpiece (sensory trick splint) [61], or muscle afferent block therapy [2,20]; (4) overseas resident, for whom regular follow-up and evaluation were not possible; and (5) common BoNT contraindications such as systemic neuromuscular junction disorders (myasthenia gravis, Lambert-Eaton syndrome, and amyotrophic lateral sclerosis), current or possible pregnancy, and lactation.

The patients were divided into six groups according to the six subtypes of OMD (jaw closing, tongue, jaw opening, jaw deviation, jaw protrusion, and lip dystonia) based on the direction of abnormal jaw movements (Table 1). If two or more subtypes coexisted in a patient, they were classified as having the most severe subtype [8,9,11]. 

Four hundred and eight patients with OMD were evaluated using the OMDRS (Appendix A) at baseline and endpoint or in stable condition. The endpoint was the time when the patient was satisfied with the therapeutic effect and BoNT therapy was completed. This means that the symptoms had subsided, and the BoNT injection was discontinued. Evaluation using the OMDRS was conducted one month after the last injection. In some patients, the symptoms had subsided and were being followed-up, but when the effect of BoNT wore off a little, they resumed injections. Patients in such stable condition on regular or additional BoNT injections were assessed by OMDRS one month after the last injection. Before the rating, the maximum occlusal force, maximum mouth opening, protrusion, and lateral movements, as well as protrusion or deviation of the tongue or lip were measured according to the subtype [9]. The maximum occlusal force was measured bilaterally on the molars three times using an occlusal force meter (GM10; Nagano Keiki Co., Tokyo, Japan) [9,39]. The patients were requested to speak or chew according to the video examination protocol, and the subsequent deviation was carefully rated. Regarding involuntary mouth closing (clenching), the patients were asked to close the mouth maximally and forcefully. Subsequently, the patients were asked the following question: “What is the percent force exerted when you close mouth involuntarily compared to the maximum bite force you have just tried?” [9].

The patients were interviewed to rate the severity, disability, and pain subscales. Questionnaires were then administered to the patients. The Japanese version of the OMDRS was used for 405 Japanese patients, whereas three international patients completed the English version of the OMDRS (Appendix A). 

Improvement was calculated as the rate of decrease in the total OMDRS score. A partial responder was defined as a patient with less than 30% improvement. 

Patients received an explanation of the treatment plan and provided written informed consent. This study was performed in accordance with the Declaration of Helsinki after approval from the institutional review board and ethics committee of Kyoto Medical Center (15-031).

### 5.2. BoNT Therapy

Before BoNT therapy was initiated, 3–5 mL of 0.5% lidocaine (Xylocaine; Sandoz K.K. Tokyo, Japan) was injected into the muscles with dystonia contracture to rule out patients whose symptoms were not caused by muscle tension [13,14,39]. Changes in involuntary movements and other symptoms were carefully examined after the lidocaine injection. If patients showed no changes in symptoms, other treatments were considered and no BoNT injections were performed. If patients showed improvement in symptoms under the effects of the local anesthetic, BoNT injections were administered [13,14]. 

BoNT injection methods have already been reported and discussed in detail for the lateral pterygoid [13,14,18,37], medial pterygoid [13,14,37], and tongue muscles [13,14,19]. BoNT (onabotulinumtoxinA; BOTOX^®^; Allergan, Irvine, CA, USA, AbbVie, North Chicago, IL, USA) was reconstituted with isotonic sodium chloride solution to reach a concentration of 2.5–5 units/0.1 mL. A disposable hypodermic needle electrode (TECA MyoJect Luer Lock, 37 mm × 25 G, 50 mm × 25 G; Natus Manufacturing Limited, Galway Gort, Ireland) was inserted into the target muscles. Correct placement of the needle electrode was confirmed under EMG guidance using an EMG instrument (Neuropack n1, MEM-8301, Nihon Kohden, Tokyo, Japan). After aspiration, BoNT was injected into the target muscles.

To prevent masticatory disturbance due to an excessively reduced bite force, the maximum occlusal force was measured on the bilateral molars using an occlusal force meter [13,14,39]. The muscles and doses of BoNT were individually determined for each patient based on their symptoms or occlusal force. The injections were continued until the patient was satisfied with the therapeutic effect and the injection was completed. The injection interval was 3–6 months depending on the patient’s symptoms and requests.

### 5.3. Statistical Analysis

All statistical analyses were performed using the statistical software package, Statistical Package for the Social Sciences for Windows (version 24.0; SPSS Japan, Tokyo, Japan). The null hypothesis was rejected at the 5% significance level (*p* < 0.05). Differences among the six subtypes were statistically compared using one-way analysis of variance. The Bonferroni method was used as a post hoc test when analysis of variance revealed significant differences. Fisher’s exact test and unpaired *t*-tests were used to assess the statistical significance of the differences in the distributions.

## Figures and Tables

**Figure 1 toxins-14-00656-f001:**
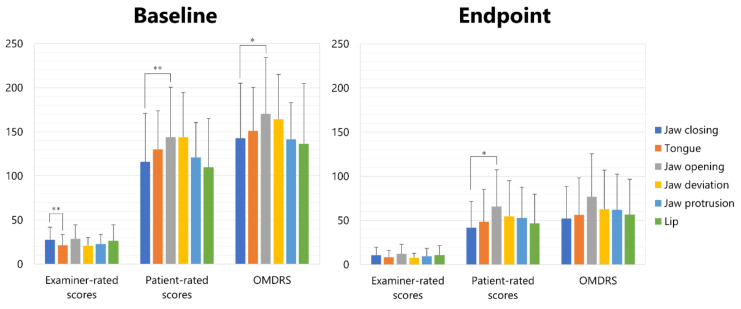
Scores of examiner-rated scale and patient-rated questionnaire scores of each subtype of OMD at baseline and endpoint. * *p* < 0.05, ** *p* < 0.01.

**Figure 2 toxins-14-00656-f002:**
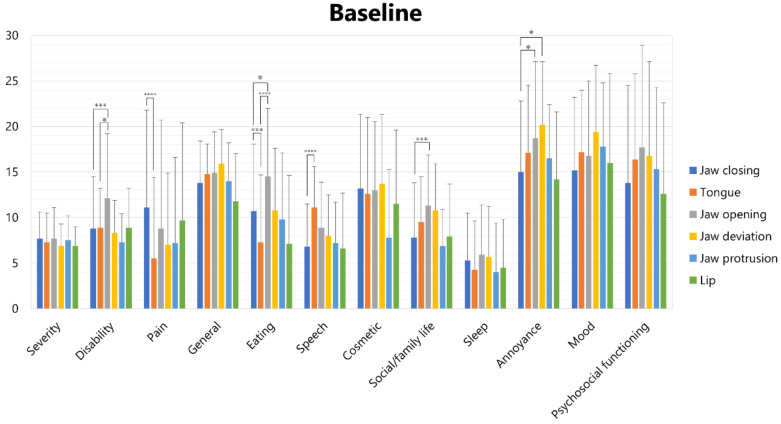
Scores of subscales of OMDRS in each subtype of OMD at baseline. * *p* < 0.05, *** *p* < 0.005, **** *p* < 0.001.

**Figure 3 toxins-14-00656-f003:**
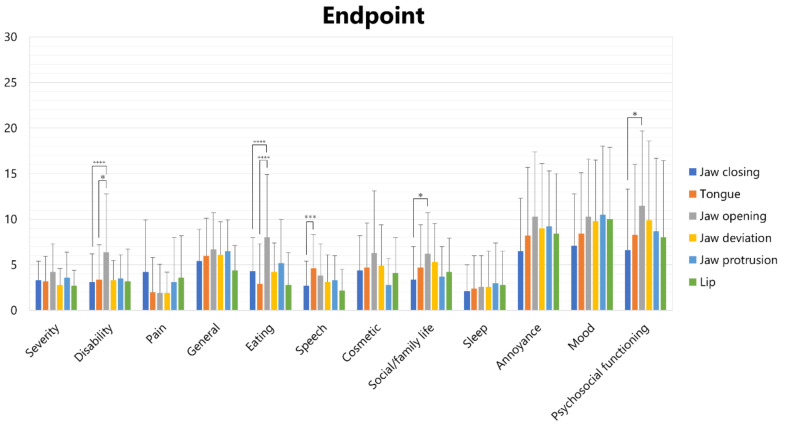
Scores of subscales of OMDRS in each subtype of OMD at the endpoint. * *p* < 0.05, *** *p* < 0.005, **** *p* < 0.001.

**Figure 4 toxins-14-00656-f004:**
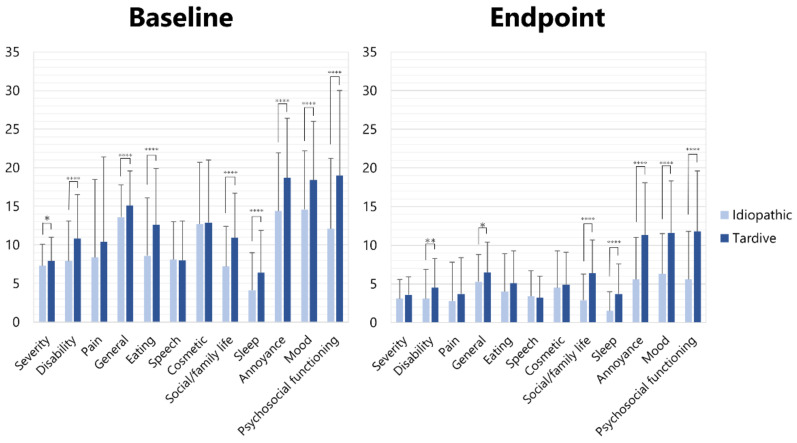
Scores of subscales of OMDRS in idiopathic and tardive cases at baseline and endpoint. * *p* < 0.05, ** *p* < 0.01, ***** p* < 0.001.

**Table 1 toxins-14-00656-t001:** Demographic characteristics for each subtype of OMD.

Subtypes	Jaw Closing	Tongue	Jaw Opening	Jaw Deviation	Jaw Protrusion	Lip	Total
No. of patients [N]	223	86	50	23	13	13	408
Age (years) [mean (SD)] (range)	53.8 (15.6) (18–95)	48.3 (14.3) (24–86)	52.3 (18.1) (19–86)	51.5 (15.7) (26–81)	49.2 (11.4) (21–63)	48.7 (11.8) (37–68)	52.0 (15.6) (18–95)
Sex (women, men) [N (%)]	155 (69.5), 68(30.5)	51 (59.3), 35 (40.7)	26 (51.0), 24 (47.1)	14 (60.9), 9 (39.1)	7 (53.8), 6 (46.2)	10 (76.9), 3 (23.1)	262 (64.2), 146 (35.8)
Duration (months) [mean (SD)] (range)	51.5 (64.6) (1–276)	39.0 (68.5) (1–180)	48.7 (91.5) (1–180)	41.0 (52.3) (2–228)	26.0 (22.1) (3–60)	56.8 (47.3) (4–156)	47.3 (67.3) (1–276)
Tardive dystonia [N (%)]	101 (45.3)	27 (31.4)	23 (45.1)	7 (30.4)	4 (28.6)	6 (46.2)	168 (41.2)
Other dystonia [N (%)]	41 (18.4)	7 (8.1)	17 (34.0)	4 (17.4)	1 (7.7)	3 (23.1)	73 (17.9)
Cervical dystonia	28 (12.6)	1 (1.2)	9 (18.0)	2 (8.7)	1 (7.7)	2 (15.4)	43 (10.5)
Blepharospasm	16 (7.2)	2 (2.3)	4 (8.0)	2 (8.7)	0	1 (7.7)	25 (6.1)
Writer’s cramp	3 (1.3)	2 (2.3)	2 (4.0)	0	0	0	7 (1.7)
Upper limb dystonia	2 (0.9)	1 (1.2)	1 (2.0)	1 (4.3)	0	0	5 (1.2)
Lower limb dystonia	2 (0.9)	0	1 (2.0)	0	0	0	3 (0.7)
Spasmodic dysphonia	1 (0.4)	0	1 (2.0)	0	0	1 (7.7)	3 (0.7)
Embouchure dystonia	1 (0.4)	1 (1.2)	0	0	0	0	2 (0.5)

**Table 2 toxins-14-00656-t002:** Results of BoNT injection for each subtype of OMD.

	Jaw Closing	Tongue	Jaw Opening	Jaw Deviation	Jaw Protrusion	Lip	Total
No. of patients [N]	223	86	50	23	13	13	408
No. of BoNT injection [mean (SD)], (range)	4.7 (4.4) (2–22)	6.4 (5.6) (3–27)	6.1 (5.2) (2–25)	6.4 (6.1) (3–25)	4.2 (3.1) (2–9)	5.8 (5.7) (2–17)	5.4 (5.0) (2–27)
No. of injected muscles [mean (SD)], (range)	3.9 (1.9) (1–12)	3.1 (1.8) (2–12)	3.8 (2.2) (2–10)	3.8 (2.4) (1–6)	3.3 (1.7) (2–6)	4.4 (1.9) (1–8)	3.7 (2.0) (1–12)
Target muscles [N (%)]	Masseter: 192 (86.1) Temporalis: 110 (49.3) Medial pterygoid: 39 (17.5) Lateral pterygoid: 26 (11.7) Posterior digastric: 10 (4.5) Orbicularis oris: 8 (3.6) Mentalis: 8 (3.6) Genioglossus: 8 (3.6) Sternocleidomastoid: 6 (2.7) Risorius: 5 (2.2) Zygomatic major 3 (1.3) Others: 7 (3.1)	Genioglossus: 86 (100) Lateral pterygoid: 19 (22.1) Masseter: 10 (11.6) Medial pterygoid: 3 (3.5) Orbicularis oris 3 (3.5) Temporalis: 2 (2.3) Posterior digastric: 2 (2.3) Others: 3 (3.5)	Lateral pterygoid: 48 (96) Anterior digastric: 10 (20) Posterior digastric: 8 (16) Genioglossus: 7 (14) Orbicularis oris 3 (6) Sternocleidomastoid: 3 (6) Platysma: 3 (6) Mentalis: 2 (4) Risorius: 1 (2)	Lateral pterygoid: 23 (100) Masseter: 7 (30.4)Temporalis: 4 (17.4) Risorius: 3 (13) Posterior digastric: 2 (9.5) Others: 4 (13)	Lateral pterygoid: 13 (100) Masseter: 3 (23.1) Temporalis: 3 (23.1)	Orbicularis oris: 9 (69.2) Risorius: 8 (61.5) Mentalis: 5 (38.5) Depressor labii inferioris: 3 (23.1) Masseter: 2 (15.4) Others: 3 (23.1)	Masseter: 210 (51.5) Lateral pterygoid: 129 (31.6) Temporalis: 119 (29.2) Genioglossus: 101 (24.8) Medical pterygoid: 42 (10.3) Orbicularis oris: 23 (5.6) Posterior digastric: 22 (5.4) Risorius: 17 (4.2) Mentalis: 15 (3.7)Anterior digastric: 10 (2.5) Sternocleidomastoid: 9 (2.2) Zygomatic major 3 (0.7) Platysma: 3 (0.7) Depressor labii inferioris: 3 (0.7) Others: 17 (4.2)
Improvement (%) [mean (SD)], (range)	63.3 (16.2) 17.1–98.1	66.3 (20.8) 21.4–97.8	57.2 (23.1) 14.9–98.4	65.8 (19.2) 28.7–87.1	58.8 (17.5) 35.0–80.5	60.6 (15.1) 31.9–86.6	63.1 (18.6) 14.9–98.4
Partial responder [N (%)]	3 (1.3)	4 (4.7)	6 (12.0)	1 (4.3)	0	0	14 (3.4)
Follow-up (months) [mean (SD)], (range)	27.8 (22.9) (12–98)	37.2 (29.3) (12–87)	35.6 (27.9) (12–74)	38.4 (34.9) (12–78)	23.6 (17.9) (12–32)	38.1 (37.1) (12–62)	31.8 (28.8) (12–87)

**Table 3 toxins-14-00656-t003:** OMDRS scores at baseline and endpoint.

OMDRS Subscale (Range)	Baseline	Endpoint	*p*-Value
Examiner-rated scale [mean (SD)]			
Severity (0–16)	7.5 (2.9)	3.3 (2.4)	*p* < 0.001
Disability (0–30)	9.1 (5.6)	3.6 (3.9)	*p* < 0.001
Pain (0–40)	9.3 (10.6)	3.2 (4.8)	*p* < 0.001
Total examiner-rated scores (0–86)	25.9 (14.1)	10.0 (9.1)	*p* < 0.001
Patient-rated questionnaire [mean (SD)]			
General (0–20)	14.2 (4.4)	5.7 (3.7)	*p* < 0.001
Eating (0–28)	10.3 (7.7)	4.4 (4.6)	*p* < 0.001
Speech (0–16)	8.0 (5.0)	3.3 (3.1)	*p* < 0.001
Cosmetic (0–28)	12.8 (8.0)	4.6 (4.6)	*p* < 0.001
Social/family life (0–20)	8.8 (5.7)	4.3 (4.1)	*p* < 0.001
Sleep (0–16)	5.1 (5.3)	2.4 (3.2)	*p* < 0.001
Annoyance (0–32)	16.2 (7.8)	7.8 (6.6)	*p* < 0.001
Mood (0–28)	16.2 (7.8)	8.3 (6.4)	*p* < 0.001
Psychosocial functioning (0–40)	15.0 (10.5)	8.0 (7.5)	*p* < 0.001
Total patient-rated scores (0–228)	123.9 (53.2)	48.2 (34.8)	*p* < 0.001
OMDRS (0–314)	148.9 (59.6)	57.6 (40.6)	*p* < 0.001

Improvement, calculated as the rate of decrease in the total OMDRS score, was significantly correlated (*r* = 0.938; *p* < 0.001) with subjective improvement. Post-treatment OMDRS scores showed a significant correlation (*r* = 0.231; *p* < 0.005) with the number of injected muscles and were significantly negatively correlated (*r* = −0.302; *p* < 0.001) with improvement in OMDRS scores.

## Data Availability

The raw data supporting the conclusions of this article will be made available by the authors, without undue reservation, to any qualified researcher.

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
