# Peer review of "Effects of Botulinum Toxin Therapy on Health-Related Quality of Life Evaluated by the Oromandibular Dystonia Rating Scale"

_toxins, 2022, doi:10.3390/toxins14100656_

Round 1
Reviewer 1 Report
In this study the authors investigated the effect of botulinum toxin injection in patients with oromandibular dystonia (OMD), as assessed by the OMD rating scale (OMDRS). The topic is interesting, and it has been only partially explored by previous studies. The authors investigated a large population of OMD patients. The experimental procedure is well designed and clear. Overall, this study provided novel information, that are useful for clinical practice. I have only some minor concerns or questions.
Methods:
1 The authors defined the endpoint as “the time when the patient was satisfied with the therapeutic effect and BoNT therapy was completed”. A better explanation of time points would clarify whether the observed results have been obtained at the same time point across all participants or not. This information is important also in order to allow other authors to reproduce this paradigm in future longitudinal studies investigating long term effects of botulinum toxin injection in OMD.
The following sentence “Patients in stable condition on regular BoNT injections were assessed by OMDRS 1 month after the last injection” is also not clear to me. Can the authors explain what they mean by “regular BoNT injections” or “stable condition” , in order to clarify the timing of assessment for all patients.
2 It would be useful to know what items of the OMDRS have been considered to identify the “partial responders” (i.e. Total Examiner-rated Scale Score, or only the severity scale..)
3 I wonder whether there is a significant difference between the clinical improvement, as assessed by clinicians’ evaluation, and the subjective improvement, as detected by the patients, after BoNT injections. It would be interesting to add this analysis in the manuscript.
Discussion:
It would be interesting to further comment the result that 10 out of 14 partial responders were patients with tardive dystonia. The different response to BoNT observed in patients with tardive dystonia compared to those with idiopathic OMD observed in this study may be due to the different pathophysiology of these two conditions. A discussion on this point would provide useful information for future investigations on mechanisms underlying botulinum toxin effects in dystonia, and also on the pathophysiology of idiopathic OMD.
I suggest the authors to move the “limitations and strength” paragraph at the end of the discussion (before “future directions”) in order to highlight the “results of this study” paragraph. Accordingly, to highlight the main message of the manuscript, I suggest to shorten the discussion paragraph.
Author Response
Methods:
1 The authors defined the endpoint as “the time when the patient was satisfied with the therapeutic effect and BoNT therapy was completed”. A better explanation of time points would clarify whether the observed results have been obtained at the same time point across all participants or not. This information is important also in order to allow other authors to reproduce this paradigm in future longitudinal studies investigating long term effects of botulinum toxin injection in OMD.
Thank you very much for your valuable comments. “the time when the patient was satisfied with the therapeutic effect and BoNT therapy was completed”. This means that the symptoms have subsided and the BoNT injection has been discontinued. Evaluation using OMDRS was conducted one month after the last injection. Therefore, the time point (one month after the last injection) was the same, naturally the date of evaluation varies from patient to patient. I added the followings in the Materials and Methods,
“This means that the symptoms had subsided, and the BoNT injection was discontinued. Evaluation using the OMDRS was conducted one month after the last injection.”
The following sentence “Patients in stable condition on regular BoNT injections were assessed by OMDRS 1 month after the last injection” is also not clear to me. Can the authors explain what they mean by “regular BoNT injections” or “stable condition”, in order to clarify the timing of assessment for all patients.
In some patients, the symptoms had subsided and were being followed-up, but when the effect of BoNT wore off a little, they resumed injections. I added the following sentences in the Materials and Methods.
“In some patients, the symptoms had subsided and were being followed-up, but when the effect of BoNT wore off a little, they resumed injections. Patients in such stable condition on regular or additional BoNT injections were assessed by OMDRS one month after the last injection.”
2 It would be useful to know what items of the OMDRS have been considered to identify the “partial responders” (i.e. Total Examiner-rated Scale Score, or only the severity scale..)
I used total OMDRS score. I have corrected the sentence as below.
“Improvement was calculated as the rate of decrease in the total OMDRS score. A partial responder was defined as a patient with less than 30% improvement.”
3 I wonder whether there is a significant difference between the clinical improvement, as assessed by clinicians’ evaluation, and the subjective improvement, as detected by the patients, after BoNT injections. It would be interesting to add this analysis in the manuscript.
Thank you for your interesting suggestion. I often ask my patients for subjective improvement when I see them. It's all recorded in the chart. Sometimes I am disappointed when the subjective improvement of a patient who has objectively improved greatly is not so good. Perhaps many clinicians have such experiences. Reasons for such discrepancies may correspond to differences in examiner-rated and patient-rated scores or differences in the effect of motor or non-motor symptoms. I analyzed the correlation between improvement and subjective improvement, differences between examiner-rated and patient-rated scores, and differences between idiopathic and tardive patients. I added following two sentences in the Result.
“Improvement, calculated as the rate of decrease in the total OMDRS score, was significantly correlated (r=0.938; p < 0.001) with subjective improvement.”
“Subjective improvement was also significantly lower in tardive patients (52.3 ± 18.8%) than in idiopathic patients (67.6 ± 17.4%, p < 0.001). Improvements in patient-rated scores were significantly lower in tardive patients (57.4 ± 17.8%) than in idiopathic patients (65.5 ± 19.1%, p < 0.001). While, improvement of examiner-rated score was lower in tardive cases (60.6 ± 20.1%) than in idiopathic cases (64.6 ± 20.3%), it was not significant (p=0.15).”
Discussion:
It would be interesting to further comment the result that 10 out of 14 partial responders were patients with tardive dystonia. The different response to BoNT observed in patients with tardive dystonia compared to those with idiopathic OMD observed in this study may be due to the different pathophysiology of these two conditions. A discussion on this point would provide useful information for future investigations on mechanisms underlying botulinum toxin effects in dystonia, and also on the pathophysiology of idiopathic OMD.
I added following sentences in the Discussion.
“However, improvement in the patient-rated score was significantly lower in tardive patients (57.4 ± 17.8%) than in idiopathic patients (65.5 ± 19.1%, p < 0.001) in this study. Moreover, scores for non-motor symptoms (annoyance, mood, and psychosocial functioning) were significantly (p < 0.001) higher in tardive cases than in idiopathic cases (Figure 4). The differences might result in low improvement in the ten patients. The possibility of non-responders was not confirmed in laboratory or clinical tests.”
I suggest the authors to move the “limitations and strength” paragraph at the end of the discussion (before “future directions”) in order to highlight the “results of this study” paragraph. Accordingly, to highlight the main message of the manuscript, I suggest to shorten the discussion paragraph.
I moved the “limitations and strengths” paragraph at the end of the discussion before “future direction”.

Reviewer 2 Report
Dear Authors,
The objective of the present study was to evaluate post-treatment changes in botulinum toxin therapy at the endpoint or in a stable status, particularly in the quality of life, in patients with oromandibular dystonia using the Oromandibular Dystonia Rating Scale.
The study was in line with the aims of the journal.
However, there are some issues that should be addressed.
Abstract
Please report in the Abstract “Health Related Quality of Life (HRQoL)” and then use HRQoL.
Introduction
- Please report in the Introduction Section “Health Related Quality of Life (HRQoL)” and then use HRQoL in all the text.
- I suggest modifying the sentence “Four hundred and eight patients (jaw closing dystonia, 223; tongue [lingual] dystonia, 86; jaw opening dystonia, 50; jaw deviation dystonia, 23; jaw protru-sion dystonia, 13; and lip [labial] dystonia, 13) were evaluated at baseline and after the end of botu-inum toxin therapy or in a stable status” to “Four hundred and eight patients (jaw closing dystonia, n=223; tongue [lingual] dystonia, n=86; jaw opening dystonia, n=50; jaw deviation dystonia, n=23; jaw protru-sion dystonia, n=13; and lip [labial] dystonia, n=13).
- “Diverse”?
- Please add some epidemiological data on oromandibular dystonia.
- In the introduction Section please report a brief introduciot about botulinum toxin therapy and its usefulness in oro-facial district (Please cite “De la Torre et al. Efficacy of Botulinum Toxin Type-A I in the Improvement of Mandibular Motion and Muscle Sensibility in Myofascial Pain TMD Subjects: A Randomized Controlled Trial. Toxins (Basel). 2022 Jun 29;14(7):441. doi: 10.3390/toxins14070441” and “Ferrillo M et al. Efficacy of rehabilitation on reducing pain in muscle-related temporomandibular disorders: A systematic review and meta-analysis of randomized controlled trials. J Back Musculoskelet Rehabil. 2022 Feb 18. doi: 10.3233/BMR-210236”).
Results
- “The mean follow-up duration was 31.8 ± 28.8 months”. Is this time the “Endpoint” (T1)? Please clarify.
Discussion
- Please report the Limitations and strengths Section at the end of the discussion Section.
Materials and Methods
- I suggest modify the Section “Patients” to “Participants”.
- “Four hundred and eight patients (262 women and 146 men; mean age, 52.0 ± 15.6 years) with OMD were enrolled in this study (Table 1).” Put these information only in the Result Section.
- Where patients where enrolled?
- When patients were enrolled?
- Is it a retrospective study? Please report this information at the beginning of the Section.
Author Response
Abstract
Please report in the Abstract “Health Related Quality of Life (HRQoL)” and then use HRQoL.
I used HRQoL as an abbreviation for Health Related Quality of Life. I think that abbreviations should generally be avoided in abstracts, however as I used HRQoL, I also decided to use abbreviations for the more repetitive OMD (oromandibular dystonia) and BoNT (botulinum neurotoxin).
Introduction
- Please report in the Introduction Section “Health Related Quality of Life (HRQoL)” and then use HRQoL in all the text.
Thank you for your comments. I used HRQoL as an abbreviation for Health Related Quality of Life in all the text.
- I suggest modifying the sentence “Four hundred and eight patients (jaw closing dystonia, 223; tongue [lingual] dystonia, 86; jaw opening dystonia, 50; jaw deviation dystonia, 23; jaw protru-sion dystonia, 13; and lip [labial] dystonia, 13) were evaluated at baseline and after the end of botu-inum toxin therapy or in a stable status” to “Four hundred and eight patients (jaw closing dystonia, n=223; tongue [lingual] dystonia, n=86; jaw opening dystonia, n=50; jaw deviation dystonia, n=23; jaw protru-sion dystonia, n=13; and lip [labial] dystonia, n=13).
I modified the sentence as you suggested.
- “Diverse”?
I used the word ‘diverse’ to mean ‘various’, but I think ‘various’ would be more appropriate in this sentence, so I changed it to ‘various’.
- Please add some epidemiological data on oromandibular dystonia.
I added the following sentence and cited the literature.
“Previously reported estimated prevalence of OMD varied from 0.1 to 9.8 per 100,000 persons [11,15.16]”
- In the introduction Section please report a brief introduciot about botulinum toxin therapy and its usefulness in oro-facial district (Please cite “De la Torre et al. Efficacy of Botulinum Toxin Type-A I in the Improvement of Mandibular Motion and Muscle Sensibility in Myofascial Pain TMD Subjects: A Randomized Controlled Trial. Toxins (Basel). 2022 Jun 29;14(7):441. doi: 10.3390/toxins14070441” and “Ferrillo M et al. Efficacy of rehabilitation on reducing pain in muscle-related temporomandibular disorders: A systematic review and meta-analysis of randomized controlled trials. J Back Musculoskelet Rehabil. 2022 Feb 18. doi: 10.3233/BMR-210236”).
Thank you for introducing me two excellent articles. I read them with great interest. I cited the articles and discussed its relation to jaw closing dystonia not in the Introduction but in the Discussion. I added following sentences.
“Meanwhile, randomized controlled trials have already been reported for bruxism or temporomandibular disorders, which have symptoms similar to jaw closing dystonia and involuntary jaw closing [66–68]. This is likely because bruxism and temporal disorders have a higher prevalence than OMD and there are many more clinicians and researchers. In future, randomized controlled trials with a higher level of evidence should be conducted for OMD.”
Results
- “The mean follow-up duration was 31.8 ± 28.8 months”. Is this time the “Endpoint” (T1)? Please clarify.
I modified the sentence as follows;
“The mean follow-up duration from the first visit to the evaluation using OMDRS for this study was 31.8 ± 28.8 months (Table 2).”
Discussion
- Please report the Limitations and strengths Section at the end of the discussion Section.
I moved the “limitations and strengths” paragraph at the end of the discussion before “future direction”.
Materials and Methods
- I suggest modify the Section “Patients” to “Participants”.
I changed "Patients" to "Participants".
- “Four hundred and eight patients (262 women and 146 men; mean age, 52.0 ± 15.6 years) with OMD were enrolled in this study (Table 1).” Put these information only in the Result Section.
- Where patients where enrolled?
- When patients were enrolled?
I modified the sentence as follows;
“Four hundred and eight patients (262 women and 146 men; mean age, 52.0 ± 15.6 years) with OMD, who visited our department from January 2016 to January 2021, were enrolled in this study (Table 1).”
- Is it a retrospective study? Please report this information at the beginning of the Section.
It is a retrospective study. I added the information in the text.

Reviewer 3 Report
Botulinum neurotoxins have been used for oromandibular dystonia (OMD). Therefore, it is not surprising that that n had a highly positive impact on patient quality of life. Overall, the manuscript provided detailed comparisons of OMDRS at baseline and endpoint after BoNT treatment. Nonetheless, the authors provide more analysis that is in line with the current practice of using BoNT to treat OMD. However, authors did not specify which serotype of BoNT they used in this manuscript. The authors used BoTox from Allergan, which is type A botulinum neurotoxin. Authors should specify that information in the manuscript.
Author Response
Comments and Suggestions for Authors
Botulinum neurotoxins have been used for oromandibular dystonia (OMD). Therefore, it is not surprising that that n had a highly positive impact on patient quality of life. Overall, the manuscript provided detailed comparisons of OMDRS at baseline and endpoint after BoNT treatment. Nonetheless, the authors provide more analysis that is in line with the current practice of using BoNT to treat OMD. However, authors did not specify which serotype of BoNT they used in this manuscript. The authors used BoTox from Allergan, which is type A botulinum neurotoxin. Authors should specify that information in the manuscript.
Thank you for your comment. I forgot to mention the BoNT serotype. I added and corrected the sentence as follows.
“BoNT (onabotulinumtoxinA; BOTOX®; Allergan, Irvine, CA, USA, AbbVie, North Chicago, IL, USA) was reconstituted with isotonic sodium chloride solution to reach a concentration of 2.5–5 units/0.1 mL.”

Reviewer 4 Report
This study addresses the effects of botulinum toxins (BoNT) on not only the motor aspects, but also the quality of life for patients with oromandibular dystonia (OMD). This was measured using the using the Oromandibular Dystonia Rating Scale (OMDRS). The OMDRS is a rating scale that measures patient and physicians’ perception of the patient’s quality of life. The investigators found that there was statistically significant improvement after BoNT in both the motor improvement and quality of life.
I offer the following comments:
1. The authors of this study used specialized equipment, such as an occlusal force meter, to track patient mouth movement force to help in the decision of where and what dosage to inject the mouth muscles with. Since this type of specialized equipment is not readily available to general neurologists, the authors should comment about the necessity of this equipment and whether anatomic palpation is sufficient to determine the target muscles.
2. Although the OMDRS assess “mood” it should specifically assess depression and anxiety, two common non-motor features of OMD.
3. What is meant by psychosocial function?
4. The overall mean improvement in the OMDRS scores was 63.1 ± 18.6%, but the investigators should specifically address why opening OMD is more difficult to treat than the other dystonias.
5. It’s hard to believe that swallowing problems occurred in only 10% of patients, and apparently only in those with lingual dystonia. This is the most feared side effect of BoNT for OMD. The authors need to address this highly unusual finding. How many patients with OMD had swallowing problems at baseline? How was swallowing assessed?
6. Likewise, dysarthria is a common problem in patients with OMD and may be exacerbated by BoNT. Yet this prominent symptom is not even discussed.
7. Another discrepancy between this study and other studies of OMD is the lack of discussion about bruxism and temporomandibular joint syndrome – two common complications of jaw-closing OMD. The authors should briefly discuss BoNT in bruxism and cite relevant studies, including Ondo WG, Simmons JH, Shahid MH, Hashem V, Hunter C, Jankovic J. Onabotulinum toxin-A injections for sleep bruxism: A double-blind, placebo-controlled study. Neurology. 2018 Feb 13;90(7):e559-e564.
8. One of the major limitations of this study is its open-label design without placebo control as the patients came to the specialized center with high expectations of improvement.
9. Was the OMDRS validated against physician’s examination?
Author Response
- The authors of this study used specialized equipment, such as an occlusal force meter, to track patient mouth movement force to help in the decision of where and what dosage to inject the mouth muscles with. Since this type of specialized equipment is not readily available to general neurologists, the authors should comment about the necessity of this equipment and whether anatomic palpation is sufficient to determine the target muscles.
Thank you for your valuable comments. The occlusal force is one of the few indices that can objectively evaluate oral function. The occlusal force is important for evaluating hyperactivity of the jaw closing muscles and for preventing excessive occlusal force reduction due to repeated BoNT injections. Various occlusal force meters are commercially available. It's not very expensive, so I think clinicians who see many patients with jaw closing OMD should have the equipment, if possible. Generally, anatomic palpation alone is not sufficient to determine the target muscles for OMD. At least, examination of oral function and evaluation of muscle activity using EMG are essential. I wrote following sentence 5.2. BoNT therapy.
“To prevent masticatory disturbance due to an excessively reduced bite force, the maximum occlusal force was measured on the bilateral molars using an occlusal force meter [13,14,39].”
Although the OMDRS assess “mood” it should specifically assess depression and anxiety, two common non-motor features of OMD.
It is as you pointed out. I discussed it in "Limitations and strengths" as follows.
“Another weakness of this study was the lack of assessment of depression or anxiety using a dedicated rating scale or specialized examination. Further randomized controlled studies evaluating the mental status in greater detail are necessary.”
What is meant by psychosocial function?
I didn't name it "psychosocial functioning". OMDRS was created by modifying CDIP-58 (Cano et al. 2004). In CDPI-58, psychosocial sequelae includes three items: annoyance, mood, and psychosocial functioning. From the contents of the questionnaire, I consider that psychosocial functioning includes items for evaluating depression and mental state in social life. I cited following literature.
Cano, S.J.; Warner, T.T.; Linacre, J.M.; Bhatia, K.P.; Thompson, A.J.; Fitzpatrick, R.; Hobart, J.C. Capturing the true burden of dystonia on patients: the Cervical Dystonia Impact Profile (CDIP-58). Neurology 2004, 63, 1629–1633. doi: 10.1212/01.wnl.0000142962.11881.26.
- The overall mean improvement in the OMDRS scores was 63.1 ± 18.6%, but the investigators should specifically address why opening OMD is more difficult to treat than the other dystonias.
I discussed it in “Results of this study” as follows,
“A significantly lower effect of BoNT therapy in jaw opening dystonia than in jaw closing dystonia has been reported [46]. It is considered to be related to the difficulty of accurate injection into the jaw opening muscles (the lateral pterygoid muscle and anterior belly of the digastric muscle) compared to that into the jaw closing muscles (the masseter, temporalis, and medial pterygoid muscles). As the author always injects BoNT under electromyographic (EMG) guidance, BoNT must be administered precisely to the lateral pterygoid or digastric muscles. Moreover, the author used a computer-aided design/computer-assisted manufacture-derived insertion guide into the lateral pterygoid muscles for patients with jaw opening who responded insufficiently using freehand insertion [37,45]. The lateral pterygoid is a muscle involved in mastication and usually has two heads: the inferior (lower) and superior (upper) [48–50As the bilateral inferior heads contract, the condyle is pulled forward and slightly downward. If the muscle is activated only on one side, the inferior jaw rotates around a vertical axis that runs through the contralateral condyle and is pulled medially to the contralateral side [48–50]. The superior and inferior heads are activated alternately during chewing, such that the inferior head contracts during mouth opening, while the superior head relaxes [48–50]. Nevertheless, the number of lateral pterygoid muscle heads remains controversial. It is commonly a two-headed muscle, but one-headed or three-headed muscles have also been reported [46,49,50]. Detailed findings of the origins and insertions of an anatomical study suggest that the lateral pterygoid muscle is a single muscle with no clear border, containing fibers in various directions, indicating that a two-head muscle pattern would be indicated by the differences in the convergences of the muscle fibers [49]. In contrast, a recent systematic review reported that the frequency of one-headed lateral pterygoid muscles ranged from 7.7% to 26.7%, two-headed muscles from 61.4% to 91.1%, and three-headed muscles from 4.0% to 35.0% [50]. The difference in efficacy of BoNT therapy between jaw closing and jaw opening dystonia may be associated with the anatomical variability of the lateral pterygoid muscle. However, further studies with larger sample sizes are warranted.”
- It’s hard to believe that swallowing problems occurred in only 10% of patients, and apparently only in those with lingual dystonia. This is the most feared side effect of BoNT for OMD. The authors need to address this highly unusual finding. How many patients with OMD had swallowing problems at baseline? How was swallowing assessed?
Thank you for your valuable comment. I noticed that I didn't mention the patient's symptoms. Only about 10% of patients with lingual dystonia experienced temporary worsening swallowing difficulties after BoNT injection. Some patients may experience a temporary difficulty in chewing, swallowing, and speaking after BoNT injection. I think this is probably not a side effect, but a temporary imbalance in muscle coordination due to the muscle relaxant effects of BoNT. These symptoms resolve spontaneously within a short period of time. The majority of side effects are thought to be related to the injection technique and are avoided by accurate knowledge of the local anatomy, precise injection procedures, and optimal dose of BoNT.
The population in the present report must differ significantly from that of neurologic clinic, which mostly treats patients with more severe dystonia secondary to Parkinson’s disease, pantothenate kinase associated neurodegeneration, Wilson’s disease, chorea-acanthocytosis, Lesch Nyhan syndrome, and Leigh syndrome, or of ischemic or hemorrhagic stroke, tumors, infarction, and brain injury. Such severe cases are also referred to our department. However, secondary and generalized cases related to neurological diseases were excluded from the analysis of this study as these patients were unlikely to have been able to precisely rate symptoms for oromandibular dystonia in isolation from other dystonic symptoms. Regarding swallowing problem, only an interview was conducted, and a specialized examination on swallowing was not performed. However, most of cases were very mild.
I added following sentences in the Result.
“The main symptoms of the patients were masticatory disturbance (n=146, 35.8%), discomfort, and cosmetic problems associated with involuntary movement (n=123, 30.1%), pain (n=78, 19.1%), dysarthria (n=68, 16.7%), and dysphagia (n=41, 10.0%). Evaluation of the OMDRS revealed that such symptoms were more prevalent (cosmetic problem [n=314, 77.0%], masticatory disturbance [n=267, 65.2%], dysarthria [n=230, 56.4%], pain [n=199, 48.8%], and dysphagia [n=159, 38.9%]).”
Moreover, I added next sentences in the Discussion.
“The chief complaints of the patients (masticatory disturbance, 35.8%; discomfort and cosmetic problems associated with involuntary movement, 30.1%; pain, 19.1%; dysarthria, 16.7%; and dysphagia, 10.0%) were more prevalent (cosmetic problems, 77.0%; masticatory disturbance, 65.2%; dysarthria, 56.4%; pain, 48.8%; and dysphagia, 38.9%) after evaluation of OMDRS. The OMDRS could be used to evaluate very mild symptoms other than the chief complaint in more detail.”
Likewise, dysarthria is a common problem in patients with OMD and may be exacerbated by BoNT. Yet this prominent symptom is not even discussed.
Please read answer to comment 4. More than 30 years ago, I also experienced some cases where symptoms worsened after BoNT injection. In the last 10 years, not a single case, at least in this study, has worsened after BoNT injection. I wrote following sentences in the Discussion.
“A complete understanding of the local anatomy of the muscles, nerves, and other tissues and accurate injection procedures are prerequisites for BoNT therapy of OMD [13,14,39]. It is important to differentially diagnose the indications for BoNT injections. If an adequate dose of BoNT can be correctly administered to the affected muscles, the symptoms will improve. Previously reported adverse effects of BoNT include temporary regional weakness, tenderness or pain at the injection site, minor discomfort during chewing, asymmetric smiles, muscle atrophy, paresthesia, and difficulty in swallowing [4,56,58,59]. The majority of these side effects are thought to be related to the injection technique and are avoided by accurate knowledge of the local anatomy, precise injection procedures, and optimal dose of BoNT [13,14].”
- Another discrepancy between this study and other studies of OMD is the lack of discussion about bruxism and temporomandibular joint syndrome – two common complications of jaw-closing OMD. The authors should briefly discuss BoNT in bruxism and cite relevant studies, including Ondo WG, Simmons JH, Shahid MH, Hashem V, Hunter C, Jankovic J. Onabotulinum toxin-A injections for sleep bruxism: A double-blind, placebo-controlled study. Neurology. 2018 Feb 13;90(7):e559-e564.
I discussed bruxism and temporomandibular disorders in the Discussion and cited the literature.
“Meanwhile, randomized controlled trials have already been reported for bruxism or temporomandibular disorders, which have symptoms similar to jaw closing dystonia and involuntary jaw closing [66–68]. This is likely because bruxism and temporal disorders have a higher prevalence than OMD and there are many more clinicians and researchers. In future, randomized controlled trials with a higher level of evidence should be conducted for OMD.”
- One of the major limitations of this study is its open-label design without placebo control as the patients came to the specialized center with high expectations of improvement.
I discussed that point as follows,
”The present study had some limitations that should be considered. An obvious limitation of this retrospective study was its open-label design without a placebo-controlled group. It is likely that the placebo effect influenced the results of this study, particularly some subscales, such as annoyance, mood, and psychosocial functioning.”
- Was the OMDRS validated against physician’s examination?
The OMDRS has been already validated and thoroughly discussed.
Yoshida, K. Development and validation of a disease-specific oromandibular dystonia rating scale (OMDRS). Front. Neurol. 2020, 11, 583177. https://doi.org/10.3389/fneur.2020.583177.

Round 2
Reviewer 2 Report
Authors modified the text according to the suggestions.
In my opinion, it is suitable for publication.
Reviewer 4 Report
I believe the authors adequately addressed the reviewers' comments.